# Anthropomorphic Robotic Hand Prosthesis Developed for Children

**DOI:** 10.3390/biomimetics9070401

**Published:** 2024-07-02

**Authors:** Pablo Medina-Coello, Blas Salvador-Domínguez, Francisco J. Badesa, José María Rodríguez Corral, Henrik Plastrotmann, Arturo Morgado-Estévez

**Affiliations:** 1Applied Robotics Research Group (TEP-940), School of Engineering, University of Cadiz, 11519 Puerto Real, Spain; blas.salvador@uca.es (B.S.-D.); josemaria.rodriguez@uca.es (J.M.R.C.); arturo.morgado@uca.es (A.M.-E.); 2Centre for Automation and Robotics (CAR) UPM-CSIC, Universidad Politecnica de Madrid (UPM), 28040 Madrid, Spain; javier.badesa@upm.es; 3Department of Electrical Engineering and Computer Science, University of Applied Science Münster, 48565 Steinfurt, Germany; henrik.plastrotmann@fh-muenster.de

**Keywords:** robotic hand, anthropomorphic design, upper limb prosthesis, child hand, kinematics, DFMA

## Abstract

The use of both hands is a common practice in everyday life. The capacity to interact with the environment is largely dependent on the ability to use both hands. A thorough review of the current state of the art reveals that commercially available prosthetic hands designed for children are very different in functionality from those developed for adults, primarily due to prosthetic hands for adults featuring a greater number of actuated joints. Many times, patients stop using their prosthetic device because they feel that it does not fit well in terms of shape and size. With the idea of solving these problems, the design of HandBot-Kid has been developed with the anthropomorphic qualities of a child between the ages of eight and twelve in mind. Fitting the features of this age range, the robotic hand has a length of 16 cm, width of 7 cm, thickness of 3.6 cm, and weight of 328 g. The prosthesis is equipped with a total of fifteen degrees of freedom (DOF), with three DOFs allocated to each finger. The concept of design for manufacturing and assembly (DFMA) has been integrated into the development process, enabling the number of parts to be optimized in order to reduce the production time and cost. The utilization of 3D printing technology in conjunction with aluminum machining enabled the manufacturing process of the robotic hand prototype to be streamlined. The flexion–extension movement of each finger exhibits a trajectory that is highly similar to that of a real human finger. The four-bar mechanism integrated into the finger design achieves a mechanical advantage (MA) of 40.33% and a fingertip pressure force of 10.23 N. Finally, HandBot-Kid was subjected to a series of studies and taxonomical tests, including Cutkosky (16 points) and Kapandji (4 points) score tests, and the functional results were compared with some commercial solutions for children mentioned in the state of the art.

## 1. Introduction

One of the most significant obstacles for engineers and researchers in the robotics field is the creation of an artificial hand that mimics the intricate functionality of the human hand. Typically, robotic hands are fashioned as grippers for performing tasks involving gripping on robotic arms. Alternatively, they may be designed with anthropomorphic features and used as robotic prostheses. Examining the notion of robotic hands with anthropomorphic features, the synopsis presented in [1,2,3,4,5,6] provides an account of the historical, contemporary, and future characteristics of this type of hand.

The human hand has the ability to execute over thirty distinct grips, as demonstrated in [7,8]. However, the majority of these grips necessitate solely one DOF in each joint. If the finger mechanism design is limited to the full opening and closing movement, it can achieve most normal grips.

A comprehensive examination of the extant literature reveals a plethora of commercial solutions for robotic hands designed for adults. However, there is a notable absence of solutions developed for children. Due to their smaller size, some models designed for adults can be used for younger patients. However, the principal drawback is the high cost involved and the risk of becoming obsolete due to rapid user growth. The functionality and number of mobility joints of prostheses designed for adults differ from those for children, as demonstrated in Figure 1.

A differential aspect is the limited functionality of the prostheses for children that are currently available on the market. The problem is that they provide low mobility for grasping everyday objects such as a glass or a toy. According to [1], children find it difficult to use the prostheses because they are unable to grasp basic objects. This is because the geometry of the object exceeds the grasping ability of the prosthesis due to limitations in finger movement.

The mobility of the thumb is fundamental to the functionality of a hand; thus, the design of the thumb must allow for different rotations and movements (flexion–extension and adduction–abduction). The thumb is crucial for a variety of grips, and a robotic hand with more DOFs will provide greater functionality. It is required that the thumb of the robotic hand have at least three DOFs to allow the finger to make the flexion–extension and abduction–adduction movements. The abduction movement is the one that allows the thumb to be placed perpendicular to the hand and to be able to perform an infinite number of grips, as shown in [9]; thus, this movement must be present in the design of the hand.

It is important to consider the aesthetic aspect of a prosthesis as well. Research suggests that even the most functional prosthetic hands are often abandoned, potentially due to designs that do not take into account the appearance and measurements of a real hand [1,10,11]. To prevent this issue, the prosthesis should resemble the appearance, shape, weight, and anthropomorphic measurements of a child between eight and twelve years old.

The results presented in [12] demonstrate that the average cost of hand prostheses is considerable, largely due to the assumed high costs of manufacturing, materials and components. These factors limit the possible use of prostheses and make them less accessible, particularly in less developed nations. The advent of 3D printing technology has made the development and manufacture of hand design more affordable, as evidenced by several significant studies, including [13,14,15,16,17]. In addition, 3D printing allows for rapid prototyping, which eases the prototyping phase of the design. Based on this technology, several recent robotic hand projects have been developed [18,19,20,21,22,23,24].

Active hands use mechanisms and actuators to recreate the movement of a real hand’s fingers. The movement of the fingers varies among these devices depending on the mechanism employed, as shown in [5,6]. Different concepts of exceptional robotic hands developed in research projects are presented below. These concepts introduce wire and bar-based mechanisms, both of which are best suited to the size constraints and limitations of a robotic hand designed to fit a child’s size.

One of the most frequently employed mechanisms in robotic hands involves the use of cables or wires for transmission. The mechanism comprises a cable or wire connected to an actuator and located internally or externally on the phalanges. This mechanism is found in the DEXMART anthropomorphic hand; see [25]. The primary issue with this kind of mechanism is that it necessitates a secondary actuator, springs, or pulleys to achieve the opposite motion.

The IH2 Azzurra prosthetic hand incorporates coil springs in its wire mechanism to facilitate finger extension. In addition, it is noteworthy that this hand has a size comparable to that of a human hand and is able to grasp a diverse range of common objects [26].

The Awiwi Hand, which belongs to the DLR Hand–Arm System, employs a wire-based mechanism with pulleys. Notably, each joint of this hand consists of a wired mechanism and pulleys that is separate from the others, allowing each actuator to operate on each joint autonomously, as described in in [27].

The movement of the finger is achieved through a set of bars, known as a four-bar linkage mechanism. This system consists of a group of interconnected bars, each connected to a phalanx. The system represents a four-bar mechanism in which all the bars are connected in a way that enables proportional movement. When one bar moves, the others also move proportionally, following a trajectory similar to that of a real finger. This bar mechanism was developed by the University of Toronto in 1999 [28]. The TBM hand designed by Northwestern University was one of the first robotic hands to integrate this mechanism, and introduced important innovations in prostheses designed for children at that time.

We conducted a search for research projects and commercial products focusing on anthropomorphic robotic hands that aim to replicate the grips and aesthetics of a human hand, observing that younger patients do not have very much choice in terms of prosthetic hand models at present. This is because research into the development of robotic hands has been directed towards prioritizing adult patients and because the restrictions of devices aimed at children are greater in terms of size and space limitations.

Among the robotic hands that have been developed in research projects, designs such as the SmartHand [29,30] and the Vanderbilt Multigrasp [31] stand out, as they aim to develop a hand that can perform a wide variety of grasps. To achieve this goal, these hands were both designed with one actuator for the index finger, one for the other fingers, and two for the thumb.

Worth further highlighting as research projects are the Hannes and DEKA robotic hands, described in [32,33], respectively, which were developed to avoid the high abandonment rate of upper limb prostheses. This is because the most advanced multi-degree-of-freedom robotic hands do not achieve the adaptability, dexterity, and complexity of a real hand because they do not take into account the anthropomorphic aspects of a real hand. The design of these hands is intended to be as close as possible to a real hand in both function and aesthetics, making them both anthropomorphic in design and as close as possible to the behavior of a real human hand.

The MyHand prosthesis manufactured by Hy5 Bionics is noteworthy for being the first commercially available hand to use an electro-hydraulic actuation system [34]. This concept originated in a European research project and has since become a commercial product. As shown in [2], 3D printing techniques are used to manufacture critical parts of the device in titanium and plastic for the less mechanically demanding parts.

The five prosthetic hand models introduced above are highly functional; Table 1 displays additional details on the weight, size, and functionality of these robotic hands.

There are several manufacturers dedicated to the production and development of robotic hands with anthropomorphic aspects; these hands are mainly intended to be used as prostheses in patients for the replacement of an upper limb. Table 2 shows the main commercial models and their characteristics. Upon examination of the table, it becomes evident that there are significant differences between the capabilities of prostheses designed for adults and those designed for children. The manufacturers listed below are the dominant players in the current market, with the highest number of sales and a reputation for innovation: Ottobock, Hy5, RSL Steeper, Touch Bionics, and Vinvent Systems.

The iLIMB prosthesis developed by Touch Bionics is the first robotic hand to be commercialized worldwide, as mentioned in [35]. The iLimb hand has fingers that act independently by using individual actuators for each finger.

Another commercial robotic hand of note is the Michelangelo prosthesis developed by the manufacturer Ottobock. This hand stands out for its anthropomorphic design and the functionality it provides, offering seven distinct grip options; see [35]. The options currently available for children are divided into active and passive prostheses. Active prostheses are those that offer the possibility of movement and do so by means of myoelectric signals or the actuation of the body. Models from the manufacturer Ottobock, such as the Electrohand 2000, MyoHand, and System Electric Hand Digital Twin, are most commonly used by young patients in Spain. This type of prosthesis only achieves pinching movements, with only one motor that moves the thumb towards the index and middle fingers, as shown in Figure 1.

There are more costly alternatives for children available on the market that provide better performance and conditions than the previously mentioned models. The market offers expensive solutions such as the Bebionic Small from Ottobock and the VINCENTyoung3+ from Vincent Systems. Both devices are designed for patients aged eight and above, with dimensions similar to those of a child’s hand between the ages of eight and twelve. However, the anthropomorphic form of these prostheses is limited because of their reduced number of joints and phalanges. As a result, these models are more similar to the prostheses developed for adults, but do not achieve the same level of mobility.

One limitation of the Bebionic small-size model is the lack of actuated abduction–adduction movement of the thumb, which requires manual movement by the patient. This limitation can hinder day-to-day operation, as certain grips require a change in thumb position that must be done manually, resulting in a loss of functionality.

The VINCENTyoung3+ model overcomes this limitation, as it is equipped with four actuators, one of which is responsible for the abduction–adduction movement. Although the actuators are shared with different fingers, this prosthesis still has the limitation of not being able to move each finger independently. Furthermore, sharing the actuators for the movement of multiple fingers restricts the force that this pediatric prosthesis can exert. According to the results published by [36], the pressure force exerted by each finger of this prosthesis is approximately 3 N. This value is significantly lower than the average pressure of a child’s finger, which according to measurements conducted in [38,39] is around 8 N for eleven-year-old children.

The most advanced prostheses in terms of functionality that can be used in children are the VINCENTevolution4, manufactured by Vincent Systems, and the iLimb Quantum, manufactured by Touch Bionics. While these prostheses are designed for adults, they can also be used for children who have outgrown other prostheses. This is because the smallest sizes of these prostheses have dimensions that remain within the percentiles of children. These prostheses addresses the limitations of other prostheses for children in terms of functionality. The thumb adduction–abduction movement is no longer manual, and each finger has an independent actuator. However, the main limitation of these prostheses is the number of DOFs, which means that the taxonomic evaluations are conditioned by their joints. For instance, the thumb has only two joints, which constrains the geometry that the prosthesis can reach when gripping cylindrical surfaces. Furthermore, the cost of these prostheses is considerably higher than others, as they are primarily intended for use by adult patients.

In summary, none of the previously presented robotic hand models meet the design requirements set out for the development of the design presented in this article. Several models of robotic hands with anthropomorphic appearance exist. However, no proposals in the literature approach the design requirements of the solution shown in Figure 2. For this reason, the development of the design and manufacture of HandBot-Kid was initiated to provide a solution to these shortcomings. Based on the fundamental aspects outlined above, the following design requirements have been defined:The robotic hand should be able to execute fundamental grips, which entails at least one DOF in each joint. To ensure that HandBot-Kid is able to grasp everyday objects, the strength of the robot finger must be equal to or greater than a child’s finger.To minimize the possibility of rejection, the robotic hand should maintain similar weight, dimensions, and aesthetics to a child’s hand in order to ensure that HandBot-Kid resembles a real child’s hand as closely as possible.The robotic hand’s thumb design should possess a minimum of three DOFs, permitting both flexion–extension and abduction–adduction actions, in order to enable HandBot-Kid to place the thumb in opposition to the other fingers.

## 2. Materials and Methods

This section presents an analysis of the design and development of HandBot-Kid. It begins by introducing the kinematics of the hand in order to provide an understanding of the internal mechanism of finger movement and the represented joints. The subsequent sections describe the design of the fingers, thumb, and palm. Finally, we discuss the methodology used in the manufacturing processes and the materials employed for the development of the demonstration model.

### 2.1. Kinematics of the Hand

The human hand has twenty three DOFs, as demonstrated in [14,34], some of which are not completely independent but maintain a certain correlation when performing common grips. With a design based on a kinematic chain of bars and a single actuator, it is possible to coordinate the movement of the finger to make it as realistic as possible, as shown in Appendix A.

The design of the finger has been divided into three phalanges (distal, middle, and proximal). Thus, each finger consists of three phalanges that are connected to each other by joints. Each of the phalangeal links represents the interphalangeal joints of a human hand, and each of these joints has one DOF, like a human hand. The joint between the finger and the hand represents the metacarpophalangeal joint, which in a human hand has two DOFs. HandBot-Kid’s metacarpophalangeal joint is limited to one DOF due to size constraints that make it impossible to add a second actuator. As a result, the design has three DOFs in each of the phalanges to represent flexion–extension movement: two DOFs for the interphalangeal joints and one DOF for the metacarpophalangeal joint, as shown in Figure 3 and Appendix A.

The thumb is positioned at right angles to the orientation of the fingers; therefore, the movements of the thumb are at right angles to the index, middle, ring, and little fingers. Abduction moves the thumb away from the fingers, making it perpendicular to the palm. Flexion moves the thumb towards the palm, while extension moves the thumb away from the palm. By placing the thumb at a right angle to the palm and rotating the fingers slightly towards the thumb, a position is produced in which the thumb pad is in opposition to the other fingers, as described in [7,16]. This opposition movement of the thumb is essential for the use of the hand in various grips and normal hand functions.

Based on how a thumb is moved, as introduced above, the design can be simplified to two phalanges (distal and proximal). The connection between the two phalanges forms the interphalangeal joint, which is similar to that of the other fingers and is responsible for the flexion–extension movement. The carpometacarpal joint, which is the joint that connects the thumb to the hand, is responsible for most of the movements. Thus, the articulation of the thumb to the hand has been developed to allow at least the adduction–abduction movement, which is essential for various types of grip. The robotic thumb design depicted in Figure 3 has a total of three degrees of freedom.

In summary, the kinematics of HandBot-Kid have been developed so as to have a DOF at each joint. Thus, the design has a total of fifteen DOFs, which enables the hand to perform various types of gripping, as demonstrated in the subsequent chapter on the taxonomy tests.

### 2.2. Mechanical Finger Design

The selection and development of the finger mechanism served as the starting point. This was followed by the anthropomorphic design of the fingers, which were created to resemble real fingers. After the finger design and actuation mechanism were determined, the remaining components of the hand were developed accordingly.

A review of different types of mechanisms commonly used in robotic hands was carried out; the results are described in the previous section. Based on the results of the review and the defined design requirements, the mechanisms that best fit this case are the four-bar linkage mechanism and the wire mechanism.

The MA of the bar and wire mechanisms was compared following a thorough investigation of relevant studies and tests. The results of this investigation showed a greater MA for the bar mechanism, as demonstrated in the study by [40]. This study utilized Matlab software to model and simulate both mechanisms, employing a quasi-static process sequence from full finger extension to the maximum flexion permitted by each mechanism. The MA was analyzed as the ratio of the output force to the input force and expressed as a percentage of the output force relative to the input force, calculated using the following equation.
(1)$$MA=100\%\cdot \frac{F_{Out}}{F_{In}}$$

The significance of MA is pivotal in deciding which mechanism to choose, as it correlates directly with the force of the actuator ($F_{In}$) and the force applied by the finger ($F_{Out}$). Consequently, the greater the MA value, the less force is required by the actuator to exert more force on the finger. It is important to consider that each mechanism has its own unique advantages and limitations; however, the four-bar linkage mechanism appears to offer a higher MA compared to the wire mechanism, as shown in [41].

The bar mechanism has a significant advantage in that it only requires one actuator to operate. This makes it an ideal solution when space is limited, as the dimensions of a robotic child’s hand are very limited. Therefore, based on the previous considerations, the best performing mechanism is the four-bar linkage mechanism.

The design process began with the development of the four-bar linkage mechanism. Figure 4a displays the initial conceptual design of the mechanism, which was simulated in Working Model 2D V6 software to ensure that it met the required functional requirements. After the mechanism concept was developed, the finger’s 3D modeling was created using CATIA V5 software. Figure 4b presents a rendering view of the CAD result of the finger design developed with the four-bar mechanism. The final design result of the manufactured and assembled finger mechanism is shown in Figure 4c.

The selected actuator was the model Micro Metal Gear-motor HP 6V manufactured by Pololu, with an extended motor shaft and a no-load current of 100 mA. The selected model is the optimal choice due to its mechanical properties and size characteristics. It uses a miniature DC motor with long-life carbon brushes and a 30:1 metal gearbox. An adapter connects the motor shaft to the M3 threaded shaft. The threaded shaft transmits the movement of the motor to the nut on the actuator bar, which in turn passes the movement on to the four-bar linkage mechanism.

The operation of the finger for the four-bar linkage mechanism is based on one of the bars coupled to the actuator, as shown in the Figure 4c. This actuator contains a threaded shaft, and the movement is transmitted to the bar through a nut. When the actuator is turned, the nut forces the bar to move linearly due to a guide that prevents the bar from rotating. The bars within the mechanism are interconnected both with each other and with the phalanges of the fingers, resulting in a synchronous movement when the actuator bar is moved. Figure 5 illustrates the synchronous flexion–extension movement between finger parts.

After developing the mechanism for internal movement, the next step was to improve the anthropomorphic design of the finger parts. As mentioned in the previous subsection, the fingers were designed with three phalanges. To assemble these phalanges, internal shafts with retaining washers were designed to connect to them and to the internal bars of the mechanism.

The anthropometric dimensions of a child’s hand between the ages of eight and twelve were considered fundamental aspects in the design of the fingers. The reason for working with this age range is that it reflects significant values in children’s growth and development, allowing for a more accurate and relevant comparison of dimensions. Table 3 presents the anthropometric measurements of a child’s hand in the aforementioned age range, with the robotic hand presented in a separate column. The last column of the table demonstrates that the dimensions of a child’s hand have been respected as much as possible in the pieces that the design allowed.

In order to streamline the manufacturing process and minimize the number of distinct components, the fingers (index, middle, ring, and little) all used the same design. These results share the same design across all fingers, with three phalanges and four inner phalanxes.

### 2.3. Mechanical Thumb Design

The challenge in designing the thumb mechanism was to replicate the movements of a human thumb as closely as possible. A human thumb has five DOFs that allow it to perform different movements, as reflected in [7]. The thumb design should include both flexion–extension and abduction–adduction movements, as established in the design requirements. Thus, the thumb design was developed to be able to perform these two main types of movement, as they are the ones that allow the fundamental grips. Figure 6 shows a time lapse rendering of the thumb in motion, which helps to illustrate the abduction and adduction positions.

Development was initiated with the consideration of the mechanism used to represent the two movements. For the flexion–extension movements, a four-bar mechanism similar to the one described for the fingers was selected. The mechanism is based on a system of internal bars that are connected to each other and to the phalanges of the thumb. When the bar that is coupled to the actuator is actuated, the remaining bars move in a synchronous movement, recreating the flexion and extension of a human finger.

For the abduction–adduction movement, a second actuator his used to provide the thumb with an additional degree of movement. This actuator is of a similar type to the one used for the rest of the fingers. The transmission of movement between the two actuators is facilitated by the use of a high-strength brass spur gear with 45 teeth, an outer diameter of 14 mm, and a width of 2 mm, which is attached to the shaft of each actuator. The action of the actuator is transmitted to another gear, which is coupled to the threaded shaft of the other actuator; in this way, the bar forces the finger to rotate. For this motion, a curved guide is incorporated that compels the finger to move at an angle to the pivot point, as can be seen in Figure 7.

In order to produce flexion–extension movement of the thumb, the Pololu’s Micro Metal Gearmotor with a 30:1 gearbox and a simple chamfered shaft similar to that of the fingers was used. For the other movement, the the same model of actuator was used, except in this case with a larger gear ratio (75:1), as this actuator requires more force to perform the turn movement.

After developing the mechanisms responsible for moving the thumb, the next step was to improve the thumb’s anthropomorphic design. As per the suggestion proposed by [43] and as explained in the previous subsection, the thumb’s design can be simplified to two phalanges, resulting in a four-bar mechanism reduced to three bars. Hence, the thumb has two phalanges and three internal bars. For the assembly of these parts, internal shafts with retaining washers similar to the ones in the fingers were utilized.

As with the fingers, the thumb’s anthropometric dimensions have been preserved to resemble those of a real child’s thumb. Table 3 displays the main measurements of the thumb. Compared with the dimensions of a real child’s finger, the results are quite similar.

### 2.4. Palm Design

HandBot-Kid has been designed in three parts to facilitate its assembly and manufacturing process. These three parts are a resin palm housing, aluminum structural skeleton, and resin dorsal housing. Figure 8 shows an exploded render view of the robotic hand, displaying the three parts into which the hand is divided and how each of them is assembled.

The aluminum structural skeleton is the core of HandBot-Kid, as it is the part to which all the other parts of the hand are attached. It is designed in the shape of the middle part of a hand, and preserves the anthropometric measurements of a child between the ages of eight and twelve, just as with the fingers.

The four similarly designed fingers are attached to this part. Each finger, although similar in shape and size, is placed in a position that replicates the real position of the fingers in a human hand. The different fingers are anchored to the aluminum structural skeleton by means of shafts with lock washers, similar to those used in the fingers. The connection between the finger and the structural skeleton is essential for the synchronized movement of the four-bar linkage mechanism. This joint works as a fixed pivot point for the movement of the other bars.

Internally, the aluminum structural skeleton has space to house the actuator and the internal bars of the finger mechanism. The recess where the actuator bar moves is designed to lock and prevent it from turning, ensuring linear motion of the bar. The finger actuators are fixed to the structure by two resin housing pieces, which are designed with the shape of the actuators in mind to ensure their attachment.

This skeleton structure is designed with internal spaces to accommodate the limit switches for the bar coupled to the actuator of each finger and the bearings placed at the top of each shaft in order to facilitate its movement and reduce friction with other parts.

As with the other parts, the resin palm and dorsal housings are designed with the shape and anthropometric measurements of a child’s hand in mind. These parts are attached to the aluminum structural skeleton to provide a realistic aesthetic and close the hand. The shapes of both are designed to resemble the front and back of the human hand. The interior of these parts contains hollow spaces that enclose and protect the various components of the hand.

The wrist is formed by two pieces that are anchored to the palm and dorsal housing cases. The electronic circuit control boards of HandBot-Kid are located inside these parts to facilitate connection with other components. The wiring for the actuators and sensors of the robotic hand is located inside these parts. A study was carried out on the distribution of the cables to facilitate their connection. The design of all hand parts was carried out while considering the easiest distribution and anchoring of the electronic parts.

Table 3 shows the dimensions of the basic parts of HandBot-Kid compared to the dimensions of a child’s hand. It can be seen that the design closely matches the dimensions of a child’s hand in most of its main parts. However, certain limitations imposed by the mechanisms, electronics, and actuators made it challenging to maintain these dimensions in some areas.

### 2.5. Manufacturing Process

The DFMA concept was adopted for the production of HandBot-Kid. This concept draws on a range of methodologies and techniques to improve the manufacturability and ease of assembly of designs, resulting in decreased production expenses. Consequently, this design strategy ensures that all parts can be manufactured while considering the assembly process, as described in [18]. This methodology should be applied at the earliest stage of design in order to identify the needs of the product. In addition, correct application of this concept leads to cost reductions, as design changes become more costly and difficult to implement as the design phases progress.

The implementation of DFMA in hand design means designing a product that can be manufactured and assembled in a way that results in cost and time reductions in the hand design phase. The design of each component was carefully considered to ensure that it can be manufactured using available machines and technologies in compliance with the DFM concept. The parts are joined using a cold metal welding process or by bolts with nuts, taking into account the DFA concept in their design to facilitate the assembly process.

The materials used to manufacture robotic hands have evolved with technological advances in various manufacturing processes. The initial passive prostheses were constructed from wood; see [4]. Nowadays, various materials are utilized, as outlined in [44], including metals, polymers, and silicone. However, no single material provides all of the cost, mechanical, and physical properties required for the development of a robotic hand, making it necessary to utilize various materials as required by each component constituting the hand.

When selecting materials for a part, it is crucial to consider various requirements, including cost, functionality, weight, and machinability. In the manufacturing process of a robotic hand, it is essential to ensure that the weight falls within the anthropometric limits of the limb being replicated. Metals are often compared to polymers due to the weight disparity between the two materials. To meet this requirement, light metals are suitable. As result, aluminum is the most commonly used metal in the development of robotic hands, as shown in [45].

To meet the anthropomorphic appearance and weight requirements of HandBot-Kid, it was decided to manufacture the parts from aluminum or UV resin. During the design and development of each part, the material was chosen based on the requirements it would be subjected to. As a result, aluminum was used to manufacture the finger parts, thumb parts, and structural skeleton, as the mechanical requirements of these components are much higher and they are functional parts. The parts used for protection or the aesthetics of the hand were made from UV resin.

According to [44,45], titanium and aluminum are the most commonly used metals in prosthetic design. For creating a functional prototype of HandBot-Kid, aluminum satisfies the required mechanical specifications and comes at a lower cost than titanium. The aluminum used to make the parts was alloy 2024. This type of aluminum offers low weight, high corrosion resistance, good mechanical properties, and a good finish.

The aluminum parts were produced using a CNC machine from the manufacturer Alarsis, model FRH170 ATC, which was produced in Murcia, Spain. This device has been designed to offer high accuracy and reliability in machining operations. It has a tolerance capability of ±0.005 mm and good production repeatability with minimal variation. The machine enables various types of operations; in our case, milling, cutting, and drilling procedures were employed in the manufacture of the required parts. The milling cutters we utilized were part of the FCR collection, which is manufactured by the same company as the machine and is specifically designed for aluminum milling.

The CNC machine is restricted to operating in three Cartesian coordinate axes, and there is a limitation on the length of cutting tools for machining aluminum. Therefore, the design of the fingers and thumb was made possible by using the DFMA concept. To achieve this, each of the phalanges is divided into two pieces along its central axis of symmetry. The two parts of each phalanx are then joined together using a cold welding process.

The structural skeleton was made from aluminum because it needs to be strong, as it is the part to which all the other parts that make up the hand are anchored. The use of aluminum in its manufacture ensures that the part has the necessary mechanical properties and that its weight is reduced.

The housing parts of the hand were made from UV resin using the 3D SLA printing technique. The printer employed for this technique was the Formlabs Form 3+. One of the principal advantages of this 3D printer is its high printing resolution, with a layer resolution of 0.025 mm and a dimensional accuracy of ±0.2 mm. The resin utilized for the pieces is known as standard resin. This resin is formulated for 3D printing intricate designs, prototypes, and general-purpose components that are not exposed to substantial loads. This material is usually used for the nonfunctional parts of prostheses in accordance with [46], as it does not have the mechanical properties to withstand large loads or stresses.

The wrist and forearm components were fabricated using PLA. These parts encase the electronic control, but are still susceptible to future adjustments. They were produced from PLA due to its cost effectiveness and industry-wide adoption as a prototype manufacturing material. The 3D printing device employed for this task was the Creatbot F430.

Figure 9 shows an image of the final prototype, illustrating the finish of the fully assembled and manufactured robotic hand. In this figure, it is possible to examine the materials and finishes of each component of the hand.

## 3. Results and Discussion

This section presents the results of tests conducted to characterize the functioning of the hand.

### 3.1. Movement Simulation and Analysis

The initial tests involved simulating and analyzing the trajectory of a finger during flexion and extension movements. The obtained values for the index robotic finger were then compared with a simulated mechanism and a real finger. This comparison allowed for the evaluation of the correlation between the trajectories and the angles of movement of each joint.

The mechanism’s design is based on a four-bar linkage mechanism, as mentioned in the previous section. This allows the robotic finger to simulate the trajectory of a real finger when performing the flexion–extension movement. Figure 10a shows a simulation of the trajectory of the finger during a flexion–extension movement in the Working Model 2D V6 software. This figure analyzes the movement of a robotic finger compared to that of a real finger, with an added trajectory line to represent their respective paths.

Figure 10b shows various frames of the flexion–extension movement of the robotic finger. The image displays a line corresponding to the trajectory of the robotic finger along with the two trajectories shown in Figure 10a. It can be seen that the trajectory of the robotic finger almost completely coincides with the trajectories of the simulated mechanism and real human finger. This confirms that the flexion–extension movement trajectory of the robotic finger is similar to that of a human finger.

The second test was a comparative study to determine the angles of movement for each joint of the fingers between a human hand and HandBot-Kid. The results are presented in Table 4, with three columns representing the achieved values. As can be seen, the total variation between the ranges of motion is calculated with a value of 94.4%. This value indicates that the range of motion for each joint is similar to that of a human hand, allowing it to perform grips similar to those of a real human hand.

Analyzing the results obtained for the two tests we performed, it can be said that the movement of the HandBot-Kid finger is practically the same as that of a human finger.

### 3.2. Force Simulation and Testing

The primary objective of this subsection is to simulate and measure the force exerted by the actuator and finger of HandBot-Kid. The obtained values are compared with those published in [38] in order to assess the robotic hand’s strength relative to that of an eleven-year-old subject’s fingers. This enables verification and characterization of HandBot-Kid’s ability to match or surpass a child’s finger strength.

All values were measured at the tip of the distal phalanx, which is the most physically optimal part of the finger for measuring force. Furthermore, the measurements published in [38,39] were taken at the fingertip; thus, they were performed in a similar way in order to compare the values of the measurements.

In terms of strength, the design of the four-bar linkage mechanism is designed to be able to exert a force similar to that of a real finger of a child aged between eight and twelve years. To satisfy this requirement, simulation of the mechanism was conducted via Working Model 2D V6 software, as depicted in Figure 11a. The simulation method involves defining the four-bar linkage mechanism design and the anchor points. It is crucial to consider which bars are mobile and moved by the actuator and which are used as anchor points. After defining the bar system, the force value of the actuator is determined in order to simulate and conduct the required measurements accurately.

The objective of the simulation was to determine the maximum force exerted by the fingertip due to the actuator force. As input force $F_{In}$, the simulation model used the motor force $F_{Mot}=F_{In}$ = 25 N. This value was obtained from the actuator datasheet. After simulating the mechanism, it was calculated that the maximum force exerted by the fingertip was $F_{Out}$ = 9.143 N. The design efficiency of the four-bar linkage mechanism was evaluated using Equation (1), resulting in a value of MA =36.57%. The results found in [41] show that the result of the simulated mechanism has an MA value similar to those achieved by the TMB hand.

The method for measuring finger pressure force [38] introduces the concept of MVC, which is defined as the maximum of the summed forces of the four fingers (index, middle, ring, and little). This value increases approximately with child growth and determines the pressure force exerted by the fingertips.

Based on the MVC measuring results performed by [38], a similar test was carried out to compare and characterize HandBot-Kid using the values of a real hand. A real-time reading was obtained using a digital scale with Arduino, a 5 kg load cell, and the HX711 amplifier module. The positioning of HandBot-Kid and the load cell for the measurements is shown in Figure 11b.

It should be emphasized that the four fingers (index, middle, ring, and little) have the same design, dimensions, and mechanism, as mentioned in the finger design section. The results shown in Figure 12 are in accordance with the measurement of the force that the tip of the distal phalanx of the index finger is capable of exerting. These results can be extrapolated to the other fingers, as they use the same design. The results of the measurements are presented in the caption of the same figure. The obtained values for the three tests indicate that each finger can exert a maximum pressure force of over 10 N. Based on these results, when compared with the measurements for an eleven-year-old child in [38,39] it can be said that the finger pressure force of HandBot-Kid is slightly higher than that of the child’s finger.

The last measurements were taken to verify the maximum force of the actuator and calculate the MA of HandBot-Kid’s finger. These measurements were necessary because the actuator force corresponds to the $F_{In}$ of the MA formula. Therefore, taking the value of $F_{Mot}=F_{In}$ = 25.36 N, which is the maximum force of the actuator in the first test in Figure 13, the maximum finger pressure force in the first test in Figure 12 was $F_{Out}$ = 10.23 N, resulting in a value of MA =40.33%, which is higher than the simulated MA value. This is because the simulation in Working Model 2D V6 software is a mathematical approximation and not an exact representation of reality. While the results can be close to reality in ideal situations, it is important to consider the limitations and the accuracy of the software.

Comparing the obtained results with the MA values shown in [41], it can be observed that they are is accurate when compared to other mechanisms. The design of HandBot-Kid’s movement mechanism ensures that the $F_{In}$ actuators are utilized to provide a significant $F_{Out}$ which is sufficient for grasping everyday objects.

### 3.3. Grab Testing

This subsection aims to characterize the gripping capacity of HandBot-Kid. Different analyses and taxonomy evaluations were developed to verify the hand’s capacity to manipulate everyday objects. The first taxonomy test was to check whether HandBot-Kid met the first design requirement defined at the end of section two. This requirement involves ensuring that the robotic hand can execute the fundamental grips. Following this, the Cutkosky and Kapanji taxonomy tests were performed to evaluate HandBot-Kid’s grasping capability and estimate its scores on these tests.

#### 3.3.1. Fundamental Grips

Schlesinger’s classification is based on the distinction of six basic types of grips. To differentiate this classification, Schlesinger used three critical terms that serve to categorize each variety of grip type, as shown in [48]. According to [49], these critical concepts are the shape of the object, the surface of the hand. and the shape of the grip. Figure 14 represents each of these fundamental grips and HandBot-Kid performing them. The figure validates that the hand can perform the fundamental grips.

#### 3.3.2. Cutkosky Taxonomy

The publication by Cutkosky [50] provides a list of sixteen grip configurations, categorized into power and precision grips. Cutkosky’s grip taxonomy enables the evaluation of efficiency in manipulating objects with different geometries, as demonstrated in several works [8,9,49,51]. This list of grip postures was used to verify the ability of HandBot-Kid to grasp everyday objects. The results are illustrated in Figure 15. Verifying compliance with Cutkosky’s taxonomy demonstrates the grasping ability of HandBot-Kid for objects of different geometric shapes and everyday use.

#### 3.3.3. Kapandji Score Test

The scoring test defined in [52] is based on evaluating the opposition ability of the thumb. This clinical tool uses a scoring system ranging from one to ten. A score of zero indicates no opposition movement of the thumb and a score of ten indicates a movement similar to that of a real human hand, as shown in [16,51]. Figure 16 displays the results obtained, with a total score of four points. The thumb’s three DOFs allow the tip of the middle finger to be in the maximum position. Based on these test results, it can be said that HandBot-Kid meets the third design requirement by being able to perform the opposition movement of the thumb.

### 3.4. Comparative Results

If these three tests are scored based on the number of grips that the hand can perform, the first test is rated at six points, the second at sixteen points, and the third at ten points. HandBot-Kid can perform all six grips in the first test, scoring the maximum number of points. It can also perform all sixteen types of grips defined by Cutkosky in the second test, achieving the maximum score. In the Kapandji test, HandBot-Kid reaches the maximum of the middle fingertip, resulting in a score of four points.

Using this scoring system, the obtained results were studied and compared with the most common pediatric prostheses, such as the MyoHand and Bebionic small-size models, as shown in Table 5. The aim was to demonstrate and differentiate the mobility capacity of HandBot-Kid, which conventional pediatric models do not reach.

The MyoHand model is a pediatric prosthesis that only allows pinching movements due to its single actuator. Therefore, the range of movement of this prosthesis is limited to a single motion. During the first test, the prosthesis would be able to perform the palmer, tip, and hook grips. It is important to note that this prosthesis has a small opening range, which limits its gripping capacity to certain object geometries. According to the Cutkosky taxonomy, the score would be six points, as it is impossible to control the movement of each finger independently with only one actuator. The Kapandji test would result in a score of two, as it allows the tips of the index and middle fingers to be touched, but not independently.

For a prosthetic model such as the Bebionic Small, which has a greater range of motion, the scores would be as follows. For the initial test, the score would be six points, as the prosthesis can perform the six grasping positions. According to the Cutkosky taxonomy, it would receive twelve points, as it can perform most of the grips, although some fingers share actuators and cannot move each finger independently, which does not allow it to reach the maximum score. The Kapandji test score would be two points, as the phalanges of the fingers are a single piece, making it impossible to achieve the first two scores. The main problem with this prosthesis is that movement of the thumb must be done manually, which means that the finger has to be placed in the grasping position before the move is performed, increasing the time and limiting the effective mobility of using the prosthesis.

Comparing the results of these two prostheses with those obtained by HandBot-Kid, it is evident that the range of movement and functionality of the designed robotic hand surpasses that of commercially available pediatric conventional prostheses. The only aspect in which these two commercial prostheses stand out in Table 5 is in the pressure force of each finger. However, this is a factor that is normal and superior in this context, as these two models are prostheses aimed primarily at adults, who require a higher pressure force. In the case of the Myohand, the value is high because it has only one actuator, and the measurement is made on the clamping force of the three fingers that move the actuator.

## 4. Conclusions

This article introduces the design and development of a robotic hand prosthesis aimed at children aged eight to twelve. The prosthesis is based on a concept that enhances existing commercially available solutions. The concluding section examines several key features that distinguish the described robotic hand from other commercially available prostheses.

The dimensions of HandBot-Kid are 16 cm in length and 7 cm in width, which are almost similar to those of a twelve-year-old child’s hand, as shown in Table 3. The design of HandBot-Kid is anthropomorphic, resembling the dimensions, mobility, and aesthetic appearance of a child’s hand. As mentioned in the related work section, there are no similar solutions currently available in the market. While there are solutions that consider anthropomorphic aspects, they do not achieve the same mobility as the robotic hand developed in this article.

The final result successfully fulfills the predefined design specifications, resulting in independent degrees of freedom for all four fingers (index, middle, ring, and little), enabling flexion–extension movement. Regarding the thumb, the objective was to achieve three degrees of freedom. This criterion results in adequate thumb functionality capable of conducting the flexion–extension and adduction–abduction operations autonomously. HandBot-Kid’s range of motion is similar to that of a human hand, as demonstrated in Table 4. This is noteworthy because many current solutions on the market lack comparable mobility. As a result, the kinematic aspects required for HandBot-Kid to hold day-to-day items are satisfied.

Grip tests were used to assess the ability of HandBot-Kid to perform different grip configurations. These tests were developed using the Schlesinger, Cutkosky, and Kapandji classifications. The Schlesinger test demonstrated that HandBot-Kid is capable of performing the six grip positions required to perform the fundamental grips. The results of the Cutkosky and Kapandji tests demonstrated that the fingers and thumb of HandBot-Kid are sufficiently functional to enable the grasping of everyday objects.

Regarding the simulation and analysis of finger movement, it was verified that the developed internal mechanism enables a trajectory that is practically identical to that of a real finger. Following this, it was confirmed that the design behaves very similarly to a real finger during flexion and extension movements. Subsequently, an attempt was made to characterize this movement by measuring the amount of pressure force exerted. The results of the force measurements indicated that the robotic finger mechanism is capable of exerting a maximum force of 10.23 N during the initial test, which is higher than the force generated by the real finger of an eleven-year-old child, which is less than 10 N. These tests demonstrate that the robotic finger will behave similarly to a real finger in terms of movement and force.

Cost reduction is a major concern regarding commercial prostheses due to their high pricing. The model developed in this work demonstrates that it is possible to decrease manufacturing costs without compromising technical specifications. Application of the DFMA concept can be pivotal in reducing time and expenses throughout the concept development and design processes. Linked to the application of this concept and the reduced cost of the materials involved in its development, 3D printing technologies and aluminum machining have allowed for a decrease in the time and cost of the manufacturing processes.

However, it is important to acknowledge a number of observed limitations. One significant challenge for the robotic hand is the related technological and mechanical limitations, as the size of the parts restricts the dimensions of the hand. In future versions, efforts will be made to develop a compact system that integrates the electronics, actuators, and mechanical parts in order to create a new design with a smaller size. One future line of investigation on which we will work is the development of a new robotic model with reduced dimensions and adaptability to the growth of the patient.

In consideration of the electronic aspects of the control, feedback, and operation of HandBot-Kid, it should be noted that this topic is not addressed in the present article as it is the subject of a future publication. The electronic interface has been incorporated into the design of HandBot-Kid in order to facilitate the optimal development of the project.

Finally, it can be concluded that the proposed HandBot-Kid prototype has been developed, manufactured, and evaluated and has achieved the design requirements established at the beginning of the work. The use of methods such as 3D printing and aluminum machining has enabled the creation of a cost-effective prototype that is functional for the tests that we carried out. The proposed solution is considered to be effective in the field of prostheses for children, bringing about a model that improves the functionality of these devices.

## Figures and Tables

**Figure 1 biomimetics-09-00401-f001:**
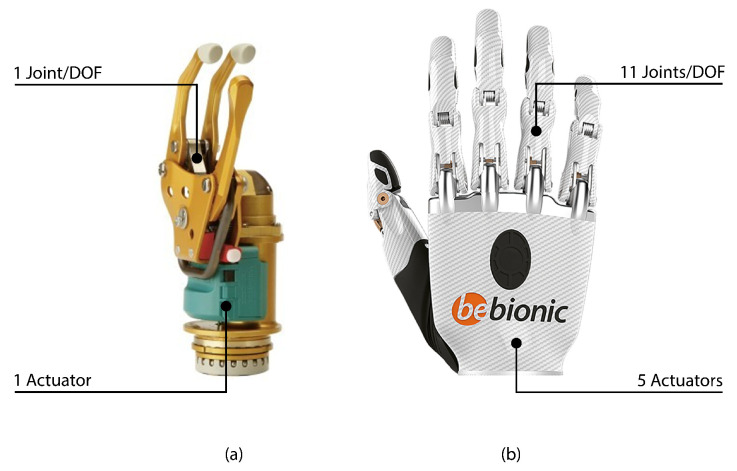
Two examples of prosthetic concepts for children currently available on the market. (**a**) The MyoHand model, manufactured by Ottobock, has limited functionality due to only having one degree of freedom. (**b**) The Bebionic model from Ottobock in its small configuration comprises five actuators, which enables it to perform a range of daily grips.

**Figure 2 biomimetics-09-00401-f002:**
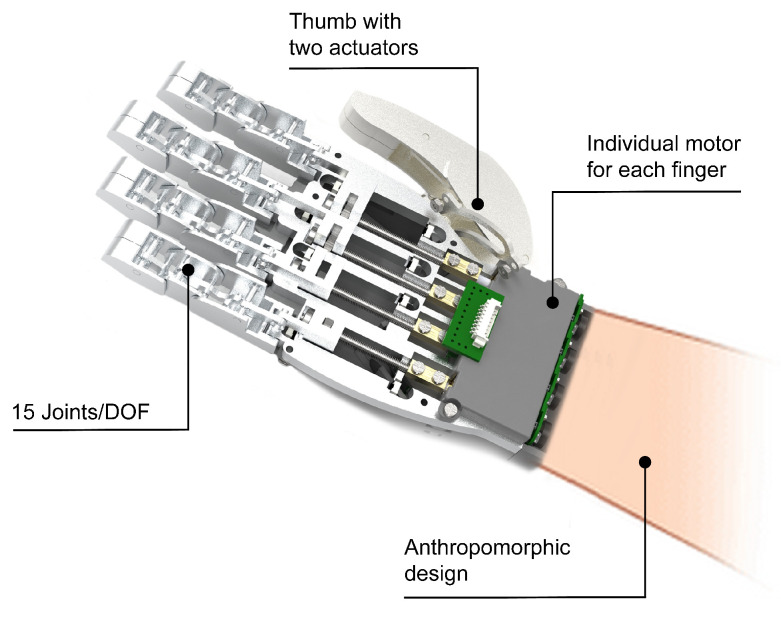
Conceptual design of HandBot-Kid. Based on the initial assumptions and design requirements established at the start of robotic hand development, efforts were made to improve upon the anthropomorphic features, aesthetics, and functionality of the models currently available on the market.

**Figure 3 biomimetics-09-00401-f003:**
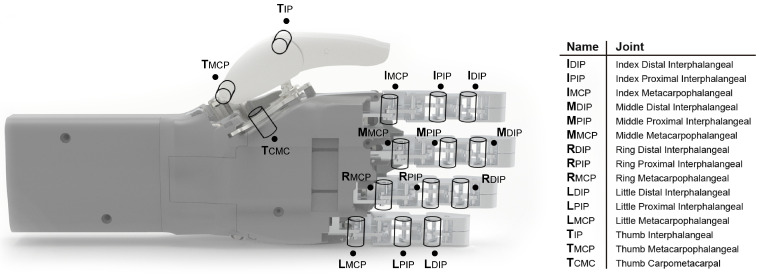
Kinematic model of HandBot-Kid. This graph illustrates the joints present in the robotic hand (fifteen joints in total), with three joints associated with each finger.

**Figure 4 biomimetics-09-00401-f004:**
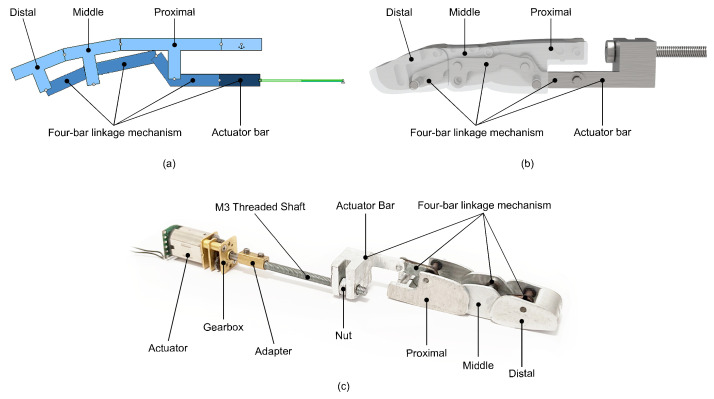
Image showing the evolution of the four-bar mechanism design process from its initial conceptualization to the final result. (**a**) Simulated view of the design and layout of the developed four-bar joint mechanism, indicating the main parts. (**b**) Rendered view of the CAD design showing how the design of the simulated mechanism was adapted to resemble a human finger. (**c**) Real image of the manufactured and assembled finger, indicating each of the different parts that compose it.

**Figure 5 biomimetics-09-00401-f005:**
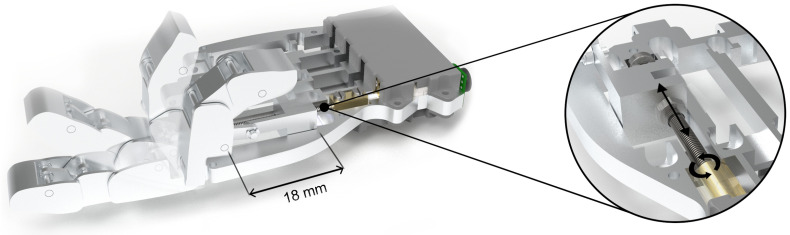
Rendered reproduction of the finger performing a flexion–extension movement, showing a time lapse representation of the finger’s movement. The actuator bar travels a distance of 18 mm from start to finish. The detail view illustrates the mechanism responsible for transforming the rotary motion of the actuator and its conversion to linear motion by a nut attached to the metric shaft. The nut transmits this movement to the actuating rod of the four-rod mechanism.

**Figure 6 biomimetics-09-00401-f006:**
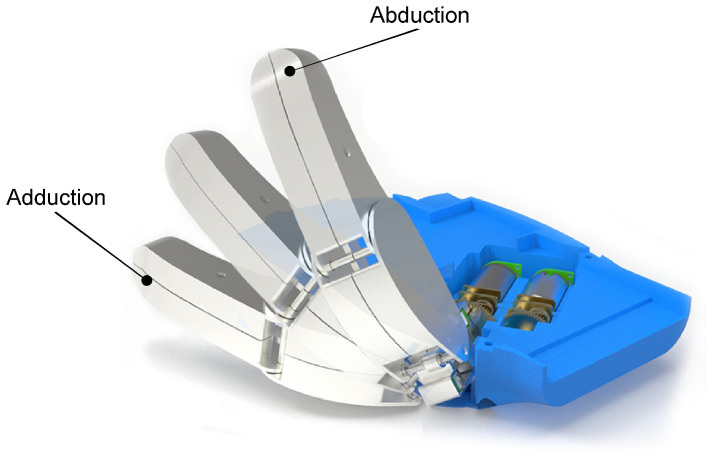
Rendered time lapse of the abduction–adduction movement.

**Figure 7 biomimetics-09-00401-f007:**
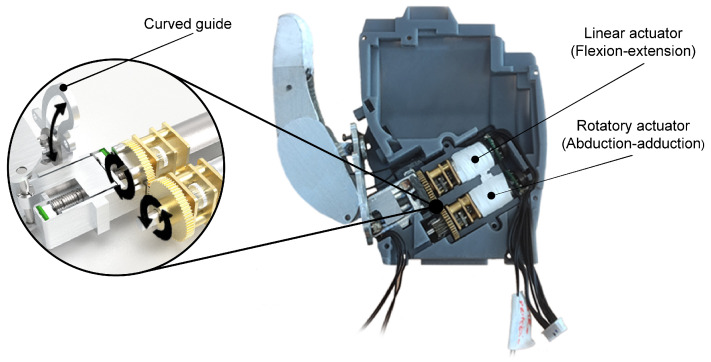
Picture showing the assembled thumb with the resin dorsal housing. A window is included to demonstrate the function of the curved guide and the transmission of motion between the rotary actuator and the linear actuator.

**Figure 8 biomimetics-09-00401-f008:**
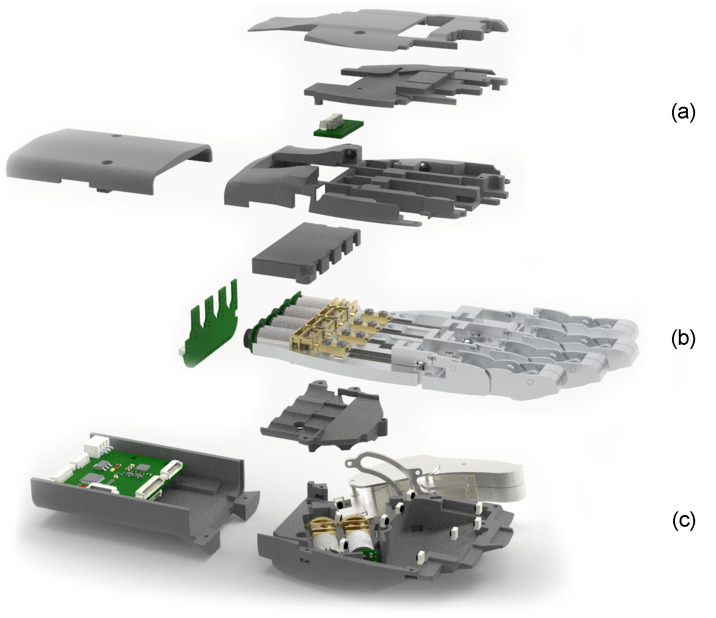
Exploded view of HandBot-Kid displaying the components comprising each part of the hand: (**a**) resin palm housing, (**b**) aluminum structural skeleton, and (**c**) resin dorsal housing.

**Figure 9 biomimetics-09-00401-f009:**
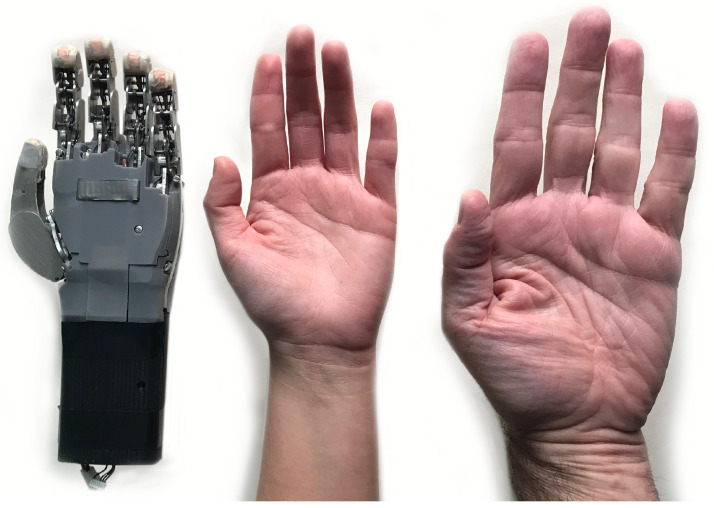
Picture showing the fully assembled and manufactured HandBot-Kid, as well as a child’s hand and an adult’s hand for comparison.

**Figure 10 biomimetics-09-00401-f010:**
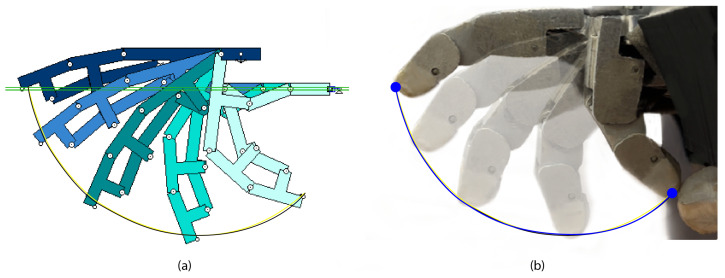
Time lapse footage illustrating the process of finger closure: (**a**) simulated finger in Working Model 2D V6 software and (**b**) robotic finger prototype. The three distinct colored trajectories represent the following: (blue) records the movement of HandBot-Kid’s index finger; (yellow) shows the trajectory trace of the simulated four-bar mechanism; and (black) represents the trajectory of a real human finger.

**Figure 11 biomimetics-09-00401-f011:**
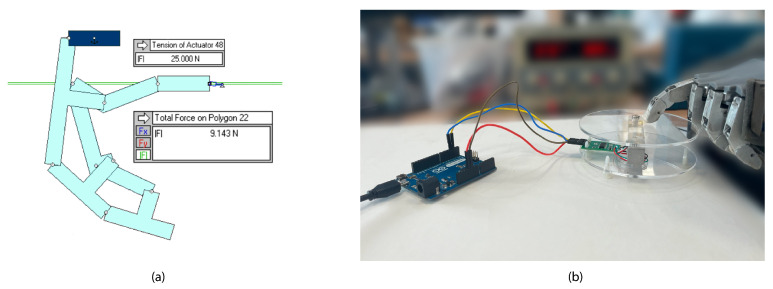
(**a**) Scene showing the simulation of the four-bar mechanism in which the maximum force is reached at the fingertip. Based on the simulation results, it can be concluded that the maximum force at the fingertip for an actuator force of $F_{In}$ = 25 N is $F_{Out}$ = 9.143 N. (**b**) This illustration shows the setup distribution and how the finger pressure force measurements were collected. The 5 kg load cell and HX711 electronic weighing module were utilized for this purpose.

**Figure 12 biomimetics-09-00401-f012:**
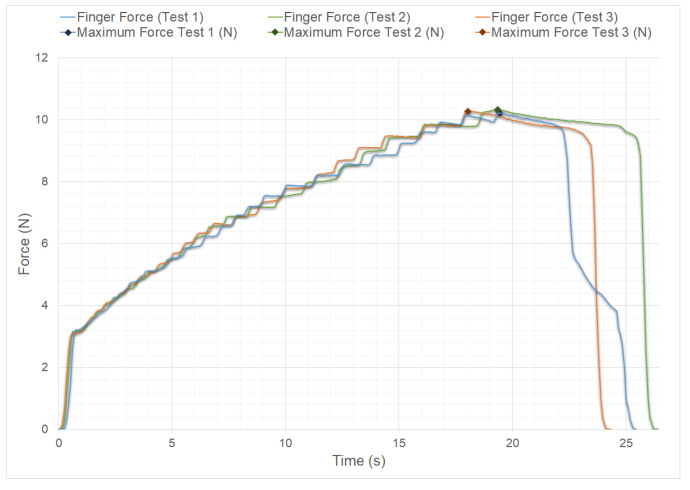
Graph displaying the pressure force values of the distal phalanx of HandBot-Kid’s finger; these force values increase over time. The graph includes three measurements, each represented by a different color, to ensure accuracy and avoid measurement errors. The maximum values achieved in each measurement were as follows: Test 1 (blue) 10.23 N; Test 2 (green) 10.33 N; Test 3 (orange) 10.28 N.

**Figure 13 biomimetics-09-00401-f013:**
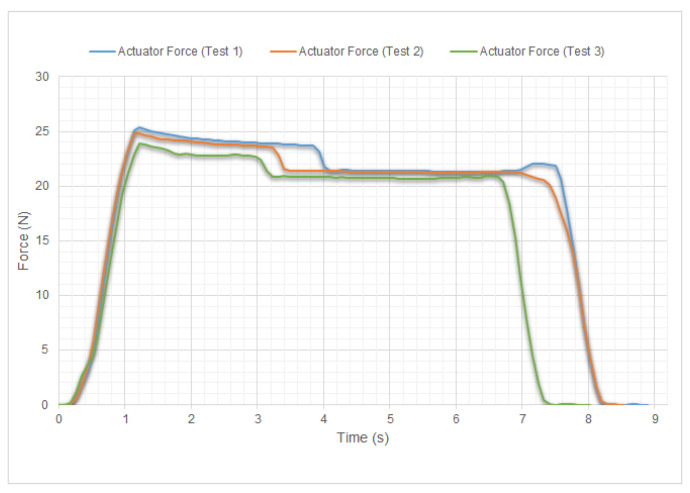
These three graphs represent measurements taken to analyze the maximum force of the actuator. The maximum force is reached rapidly and subsequently decreases gradually. According to Victor at al. [47], the actuator experiences a decline in performance and heats up when it reaches its maximum force due to saturation, resulting in a loss of power. As with the previous graph, three measurements were taken to avoid errors. The maximum values obtained for each test were as follows: Test 1 (blue) 25.36 N; Test 2 (green) 24.8 N; Test 3 (orange) 24.1 N.

**Figure 14 biomimetics-09-00401-f014:**
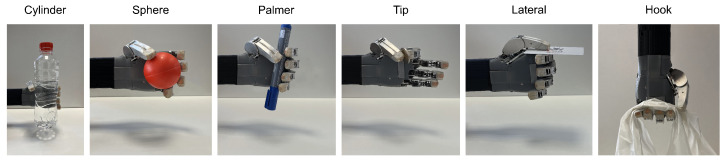
This figure illustrates the ability of HandBot-Kid to perform the fundamental grips which are commonly used to grasp everyday objects. These grips are classified into six types, as shown in [48,49].

**Figure 15 biomimetics-09-00401-f015:**
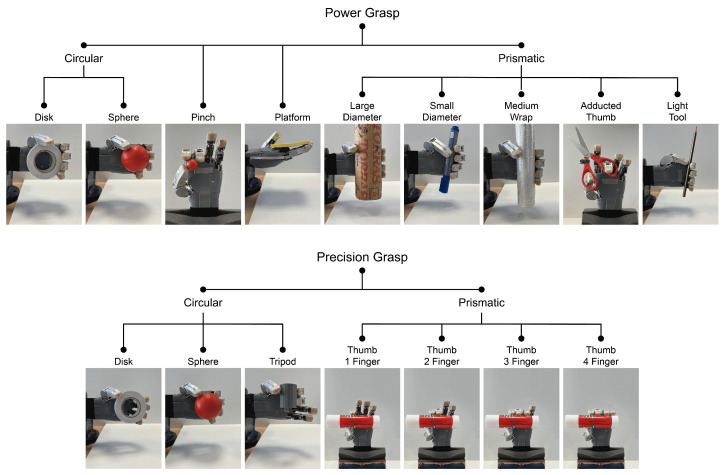
Representation of strength and precision grips according to [50]. This diagram illustrates HandBot-Kid’s ability to grasp everyday objects based on Cutkosky’s grip classification as is referenced in [8,9,49,51].

**Figure 16 biomimetics-09-00401-f016:**
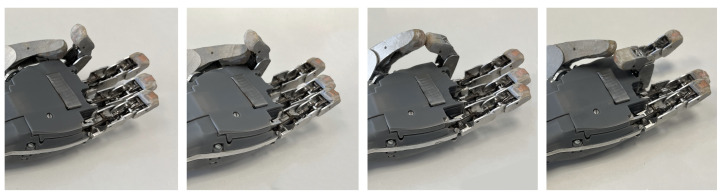
Graphics based on the opposition movement of the thumb for the Kapandji scoring test [52]. A variety of publications [8,16,51] have employed this methodology for calculating and analyzing the scores generated by this scoring system for the thumb movements of different robotic hands.

**Table 1 biomimetics-09-00401-t001:** Main characteristics of different robotic hands developed in research projects. The final column denotes the abduction–adduction motion of the thumb, with E indicating electronic movement. Some of the information provided in this table has been obtained from [2,29,30,31,32,33].

Model	Figure	Developer	Weight (g)	Size (mm)	Joints/DOF	Nº of Actuators	Active Thumb
Smarthand	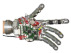	ARTS Lab (Italy)	520	195 long 86 wide 48 thick	16	4	E
Vanderbilt Multigrasp	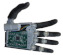	Vanderbilt University (US)	580	90 long 330 wide 75 thick	9	4	E
DEKA Arm	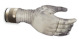	DEKA Integrated Solutions Corp. (US)	-	-	14	6	E
Hannes Hand	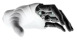	Istituto Italiano di Tecnologia (Italy)	450	196 long	14	6	E
MyHand	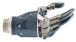	Hy5 Bionics (Norway)	575	-	7	1	-

**Table 2 biomimetics-09-00401-t002:** Characteristics of different commercial adult and child robotic hands. The final column denotes the abduction–adduction motion of the thumb, with E indicating electronic movement and M indicating manual movement. Part of the information is taken from [3,34,35,36,37].

Model	Figure	Developer	Weight (g)	Size (mm)	Joints/DOF	Nº of Actuators	Active Thumb
MyoHand *	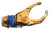	Ottobock (Germany)	350–500	178–182 long 78–82 wide	2	1	-
Michelangelo	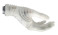	Ottobock (Germany)	420	196 long 75–80 wide 35–41 thick	6	2	E
Bebionic *	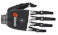	Ottobock RSL Steeper (Germany)	420–540	162–198 long 72–90 wide 42–50 thick	11	5	M
VINCENT young3+ *	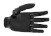	Vincent Systems (Germany)	347–436	142–185 long 65 wide	6	4	E
VINCENT evolution4 *	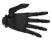	Vincent Systems (Germany)	412–556	151–215 long 72–90 wide	10	6	E
iLimb Quantum *	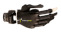	Touch Bionics (UK)	450–615	154–182 long 72–85 wide 35–41 thick	10	6	E

* The designation in the first column indicates that this product is available in different sizes for adults and adolescents. In the absence of a marking, it may be inferred that these are robotic manipulators designed for adult use.

**Table 3 biomimetics-09-00401-t003:** Dimensions of different parts of HandBot-Kid and the hand of a child between the ages of eight and twelve.

	Child Hand (cm)	HandBot-Kid (cm)	% Variation
Hand length	16.5	16	97 %
Hand width	7.5	7	93.3 %
Hand width (to the thumb)	9	9	100 %
Index length	6.4	6	93.8 %
Index diameter	1.3	1.2	92.3 %
Middle length	7	6.5	92.9 %
Middle diameter	1.35	1.2	88.9 %
Ring length	6.8	6.3	92.6 %
Ring diameter	1.3	1.2	92.3 %
Little length	5.3	5.3	100 %
Little diameter	1.2	1.2	100 %
Thumb length	4.7	4.4	93.6 %
Thumb diameter	1.5	1.4	93.3 %
		Total	94.6 %

The data in the child’s hand column were obtained from the guide created by the AIJU, a technological institute specializing in toys, children’s products, and leisure activities, based on the ergonomic design of products aimed at children, as well as from [42,43]. Among the values included in the guide, the 50th percentile values have been used, as these are the most adaptable anthropometric dimensions and satisfy the majority of normal dimensions.

**Table 4 biomimetics-09-00401-t004:** Comparison of the angles of movement of each joint in HandBot-Kid and a human hand. The second column shows the range of opening angles for each joint of HandBot-Kid. The third column shows the same values as the second in relation to the range of a real human hand. The fourth and final column displays the percentage variation between the ranges of the second and third columns, representing the similarity of each joint’s range in HandBot-Kid to that of a human hand.

	Range of Motions		
	HandBot-Kid	Human Hand	% Variation
IDIP	0–70°	0–80°	87.5%
IPIP	0–90°	0–100°	90%
IMCP	0–90°	0–90°	100%
TIP	0–70°	0–90°	88.9%
TMCP	0–50°	0–50°	100%
TCMC	0–70°	0–70°	100%
		Total	94.4%

The data for the angles of movement of each joint of a human hand were taken from the book [7]. While the table only mentions the joints of the index finger and thumb, the angle of movement of the index finger is the same for the rest of the fingers, as they all use the same design.

**Table 5 biomimetics-09-00401-t005:** Comparative analysis of the general characteristics of the developed prosthesis in comparison to two more commonly available commercial models.

	HandBot-Kid	MyoHand	Bebionic Small
Joints/DOF	15	2	11
Size (mm)	160 long 70 wide 36.5 thick	178 long 79 wide 40 thick	162 long 72 wide 42 thick
Weight (g)	328	350	420
Finger force (N)	10.23	100 *	32 *
Fundamental grips	6	3	6
Cutkosky Taxonomy	16	6	12
Kapandji score test	4	2	2

* The marked force values are identified as results that have not been measured on a single finger.

## Data Availability

The data associated with this study are available upon request by contacting the corresponding author.

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
