# Peer review of "Anthropomorphic Robotic Hand Prosthesis Developed for Children"

_biomimetics, 2024, doi:10.3390/biomimetics9070401_

Round 1

Reviewer 1 Report

Comments and Suggestions for Authors

This manuscript reports the design consideration, product fabrication, and comprehensive investigation of anthropomorphic robotic hand prosthesis for children. The developed hand prosthesis addresses the current issues and provides a superior solution to achieve functions like human hands. Overall, the manuscript is well-constructed and written, and it can be considered for publication after the authors address the following concerns.

1.       The anthropomorphic robotic hand prosthesis is designed for children between 8 and 12, can it be further reduced in size for smaller children?

2.       How is the power consumption of the anthropomorphic robotic hand prosthesis? How long can it normally be used before replacement of battery or recharge?

3.       What is the estimated cost for the developed anthropomorphic robotic hand prosthesis?

4.       The authors provide various scenarios to validate the practical functionality of the anthropomorphic robotic hand prosthesis, yet only images are provided, it is suggested to provide some videos for better illustration.

5.       The authors need to check the whole manuscript more carefully, since there are quite a few typos, format and grammar errors throughout the manuscript.

Comments on the Quality of English Language

 The authors need to check the whole manuscript more carefully, since there are quite a few typos, format and grammar errors throughout the manuscript.

Author Response

The responses to the reviewer are provided in the attached PDF document. The videos can be downloaded at the following:

Link: https://consigna.uca.es/16581

Password: 3057695

Reviewer 2 Report

Comments and Suggestions for Authors

The paper „Anthropomorphic robotic hand prosthesis developed for children” is a research paper, evidencing results of interest and estimated support  for childern with upper limb disability.

I recommend to keep on the MDPI articles’ template, meaning:  1.Introduction;   2. Materials and Methods:  3. Results and Discussion:  4. Conclusions

Please, improve the Abstract by:

-        rephrasing  at line, L1 „(...)is considered a general rule”;

-        brief justification of the targeted children age (L8) in correlation with the mentioned dimensions (L8, 9)

-        mention further research development (L19).

Please offer some relevant examples for the afirmation: „Most of the anthropomorphic robotic hands are expensive” - at L61.

Please, reconsider the structure of chapter 2. 

For example, 2.2 Background (subchapter name) does suggest both research projects and commercial solutions.   Also, Mechanisms analysis (2.1) represents background.

Consider also the MDPI template structure.

Please, consider the change of chapter 3 title - as indicated by MDPI template

At Figure 2, please correlate the number of DOF with the ones mentioned further (see Fig. 3 and  L 241, 242)

At lines 297 - 303, please detail the chosen (8 up to 12 years) children age. Further explain why Table 3 (L303) does not present data from children of 8 / 10 years old.

Please mention the actuator type and its producer / supplier.  Pololu Micro Metal is Gearmotor(L279)

Also, indicate the gear characteristics (diameter, teeth number) for the abduction-adduction movement of the thumb (L323).

I suggest to change the title of subchapter 3.5 (L383)  as it does not refer to production. It refers to prototype manufacture.

Please, mention some of the geometric precision characteristics of the parts obtained by CNC machining  (L423) and by SLA printing (L437).

Please, in the Conclusions chapter do mention some aspects regarding costs of the robotic hand developed (L660, 661).

Comments on the Quality of English Language

Check English language.

Minor changes required.

PS

Just consider the suggestion to use the noun "motion" instead of "movement" (finger motion)

Author Response

The responses to the reviewer are provided in the attached PDF document.

Reviewer 3 Report

Comments and Suggestions for Authors

The paper presents the design and development of a robotic hand prosthesis (HandBot-Kid) aimed at children aged eight to twelve. The design of the fingers, thumb and palm are shown followed by the presentation of the methodology used in the manufacturing processes and the materials employed for the development of the demonstrator. Further the operation of the robotic hand prosthesis is described and the motion limits and forces developed by the fingers are simulated and measured. Finally the performance of the Handbot-Kid is compared to that of other commercial models.

The article is interesting and written at a high scientific level. The presentation method is good and in accordance with generally accepted standards in that area. The article is of interest to researchers involved in the development of this type of hand rehabilitation systems.

Some suggestions for the improvement of the article:

·         - Can Figure 4c be enhanced so that the mechanisms responsible for motion generation become more clearly visible?

·        -  Could Figure 5 be presented more clearly?

Author Response

(The authors gave the same response as above.)

Round 2

Reviewer 2 Report

Comments and Suggestions for Authors

It is OK now